# Genetic Obesity in Pregnant *A^y^* Mice Does Not Affect Susceptibility to Obesity and Food Choice in Offspring

**DOI:** 10.3390/ijms24065610

**Published:** 2023-03-15

**Authors:** Elena Makarova, Anastasia Dubinina, Elena Denisova, Antonina Kazantseva

**Affiliations:** The Laboratory of Physiological Genetics, The Institute of Cytology and Genetics, 630090 Novosibirsk, Russia; dubinina.anastas@gmail.com (A.D.); melomande91@gmail.com (E.D.);

**Keywords:** maternal obesity, sweet and fatty diet, developmental programming, *A^y^/a* mice, food choice

## Abstract

Maternal diet and obesity (MO) may influence taste preferences and increase the susceptibility to obesity in offspring, but the impact of MO per se to these influences is poorly understood. We evaluated the influence of MO on food choice and susceptibility to obesity in offspring when mothers consumed a standard diet (SD). Mice with the *Lethal yellow* mutation (*A^y^/a*) develop obesity consuming an SD. Metabolic parameters were assessed in pregnant and lactating *A^y^/a* (obesity) and *a/a* (control) mothers. Metabolic response to the consumption of a sweet–fat diet (SFD: SD, lard, and sweet biscuits) and the choice of components of this diet were evaluated in their male and female offspring. Compared to control mothers, pregnant obese mothers had higher levels of insulin, leptin, and FGF21. MO increased food intake and liver expression of lipogenesis genes in male offspring consuming the SD. SFD consumption caused obesity development and insulin resistance, increased liver expression of glycolytic and lipogenesis genes, and affected hypothalamic expression of anorexigenic and orexigenic genes. In offspring of both sexes, MO had no effect on food choice and metabolic response to SFD intake. Therefore, when obese mothers consume a balanced diet, MO does not affect food choice and development of diet-induced obesity in offspring.

## 1. Introduction

The current high prevalence of obesity is associated with modern lifestyles: reduced physical activity, nutrition transition, and overeating [1,2]. Food choice based on taste preferences contributes to population variability in body weight [3]. The preference for fatty and sweet high-calorie foods favors overeating, and overeating induces the development of obesity [4]; in turn, the rate and degree of development of obesity depend on the composition of the diet consumed [5,6]. 

Taste preferences and the susceptibility to obesity in an obesogenic environment depend on the genotype [7,8] and the conditions of early life of individuals [9]. It has been shown that in utero environments have a significant effect on the predisposition of offspring to develop obesity later in life [10], and this effect can significantly differ in offspring of different sexes [11,12]. Maternal obesity and a maternal high-calorie diet during pregnancy and lactation were shown to increase the risk of obesity development in offspring [13,14] and to affect offspring calorie intake and taste preferences [15,16,17].

Obesity develops as a result of imbalance between energy intake and expenditure. It is assumed that exposure to an obesogenic environment during the perinatal period affects early development of the hypothalamus and disrupts the proper formation of feeding regulatory pathways [18]. Maternal overeating and diet-induced obesity have been shown to have a delayed effect on the expression of orexigenic and anorexigenic [19] hypothalamic neuropeptides participating in the maintenance of energy homeostasis in offspring.

The programming influence of the maternal environment is associated with changes in the hormonal and metabolic composition of maternal blood during pregnancy [14,20]. Elevated levels of leptin and insulin in the blood, hyperglycemia, and dyslipidemia can lead to epigenetic changes in placentas and fetuses and thus program further offspring development [21]. At the same time, the diet consumed before and after conception significantly affects the biochemical composition of maternal blood [22] and thus can also have a programming effect on the development of offspring either directly or through influence on the degree of maternal obesity. 

Most of the current research on developmental programming is based on laboratory models of diet-induced obesity caused by the consumption of high-calorie diets of various compositions [14]. In this kind of research, it is difficult to separate the programming influence of maternal diet from the influence of those metabolic disorders in mothers that are directly caused by obesity. However, in order to develop methods for correction of offspring early development, it is necessary to understand the programming effect of maternal obesity itself, even when the mother consumes a balanced, healthy diet. It is not known whether maternal obesity *per se* influences the offspring’s taste preferences, whether it increases the ability to develop obesity in offspring, and to what extent these maternal influences depend on the sex of the offspring. In order to reveal the programming effect of metabolic changes caused directly by maternal obesity, not by diet, it is possible to use genetic models of obesity that develop when consuming a standard diet.

In mice, the mutation *Lethal yellow* at the agouti locus (*A*^y^) causes ectopic overexpression of the *agouti* gene [23]. Heterozygous *A^y^/a* C57Bl mice have yellow coat color and develop obesity and non-insulin-dependent diabetes with age [24] due to ectopic expression of the *agouti* gene in the hypothalamus, which evokes chronic blockage of melanocortin receptors (MCRs) by the agouti protein [25]. In reciprocal crossings of *A^y^/a* x *a/a* and *a/a* x *A^y^/a*, *a/a* and *A^y^/a* offspring are born in a 1:1 ratio, which makes it possible to assess the effect of maternal obesity *per se* on the metabolic phenotype of offspring with a normal metabolism (*a/a*). The aim of this work was to assess the impact of maternal obesity during pregnancy and lactation on food choice and susceptibility to diet-induced obesity in offspring of different sexes. We evaluated the hormonal and metabolic blood parameters during pregnancy and lactation in females of *A^y^/a* (obesity) and *a/a* (control) genotypes, and assessed the effect of maternal obesity on metabolic parameters and the expression of liver genes involved in carbohydrate and lipid metabolism and genes encoding orexigenic and anorexigenic neuropeptides in the hypothalamus in offspring of different sexes when kept on a standard or obesogenic diet.

## 2. Results

### 2.1. Metabolic Characteristics of Mothers during Pregnancy and Lactation

*A^y^/a* females were heavier than control females by 45.5% at the time of mating (Table 1), by 20% at the end of pregnancy, and only by 12.7% at day 10 of lactation. The fat mass in *A^y^/a* females was 6.3 times higher than in the control females at the time of mating, and 3.2 times higher at the end of pregnancy. Placental weight in obese females was lower than in control females, and fetal weight at the end of pregnancy and newborn weight were also lower (Table 1); however, on postpartum day 10 (PPD10), pups born to control and obese mothers did not differ in weight. The expressions of the *Fgf21* and insulin-like growth factor 1 (*Igf1*) genes were evaluated in the pup liver; there were no differences in expression of these genes in pups born to control and obese mothers (*Fgf21*: 0.71 ± 0.18 arbitrary units (AU), n = 23, and 0.61 ± 0.07 AU, n = 22, for offspring of *a/a* and *A^y^/a* mothers, respectively; *Igf1*: 1.1 ± 0.11 AU, n = 22, and 1.15 ± 0.11 AU, n = 22, for offspring of *a/a* and *A^y^/a* mothers, respectively). 

Plasma concentrations of glucose and triglycerides were maintained at the same level during pregnancy and lactation and did not differ in control and obese female mice (Figure 1). Cholesterol concentrations decreased during pregnancy, and then increased by day 10 of lactation (*p* < 0.000 *, two-way ANOVA, “day”). Obese females differed from control females in the dynamics of changes in cholesterol levels (*p* < 0.05, two-way ANOVA, “day” * “genotype”): the average blood cholesterol level in obese females was higher at pregnancy day 5 (PD5) and lower at PPD10 than in control females (Figure 1).

During pregnancy and lactation, plasma insulin concentrations changed differently in control and obese females (*p* < 0.001, two-way ANOVA, “day” * “genotype”): insulin level increased during pregnancy, then did not change significantly during lactation in control females, and in obese females, it was more than four times higher than in control females in early pregnancy, and then decreased (Figure 1). There were no significant differences in plasma insulin concentrations at the end of pregnancy and during lactation between obese and control females. Leptin levels were higher in lactating females than in pregnant ones and did not differ between control and obese females (Figure 1). These paradoxical invert differences in leptin plasma concentrations between pregnancy and lactation may be explained by the fact that concentrations of free unbound leptin were measured, whereas most leptin is bound with the soluble extracellular domain of the leptin receptor in late pregnancy [26]. Analysis of plasma leptin levels only during pregnancy showed that leptin concentrations increased during pregnancy (*p* < 0.01, ANOVA), and this increase was higher in obese females than in control ones (“day” * “genotype”, *p* < 0.05, ANOVA). The level of FGF21 in the blood increased during pregnancy, then decreased at PPD10 (*p* < 0.000 *, ANOVA) and was higher in obese females (*p* < 0.05, ANOVA).

Thus, the maximum differences between *a/a* and *A^y^/a* females in terms of weight, fat content, and blood insulin level were observed in early pregnancy and decreased towards the end of pregnancy. In the course of pregnancy, adiposity and insulin levels increased in control females and decreased in obese females. Differences in blood hormonal and metabolic characteristics between *a/a* and *A^y^/a* females were observed only in pregnancy and disappeared in lactation.

### 2.2. Metabolic Characteristics of the Offspring of Control (a/a) and Obese (A^y^/a) Mothers after Weaning

Genetic maternal obesity had no significant effect on body weight (BW) changes in both male and female offspring that consumed an SD. SFD consumption evoked obesity, and maternal obesity did not influence weight gain during SFD consumption in male or female offspring (Figure 2A). Both males and females consumed more energy with the SFD compared to consuming the SD (Figure 2B). Maternal obesity did not influence energy intake when offspring consumed the SFD and increased intake of the SD only in males (Figure 2, repeated measures ANOVA, males, SD, factor “MG”, *p* < 0.05).

Males and females differed in their taste preferences (*p* < 0.01, three-way ANOVA, “sex” * “type of food”). Females consumed almost two times less standard chow than males and did not differ from males in the amount of consumed biscuits and fat. Females showed a preference for biscuits and consumed twice as many biscuits compared to pellets of standard chow or lard (Figure 3A). In males, the consumption of pellets of standard chow and biscuits did not differ, and they consumed less fat. The contribution of each of the components of the diet to total energy intake is shown in Figure 3B. The share of energy consumed with the SD was half that with each of the other components of the diet, while the share of energy consumed with lard and biscuits did not differ from each other. The share of energy consumed with the SD in females was lower than in males. Maternal obesity did not have a statistically significant effect on the choice of diet components.

SFD intake significantly increased visceral and subcutaneous fat mass in males and females, but had no effect on liver weight, and therefore relative liver weight in SFD-fed mice was proportionally reduced in both sexes. The absolute and relative mass of visceral fat did not depend on sex, but the accumulation of subcutaneous fat was higher in females (Table 2). Maternal obesity had no effect on fat masses and liver weight in the offspring. 

SFD consumption and obesity development were accompanied by increases in the plasma concentrations of glucose, insulin, and FGF21 in mice of both sexes, regardless of maternal obesity (Table 3). Separate analyses for males and females showed that only in females, regardless of maternal genotype, SFD intake increased liver glycogen stores (*p* < 0.01, two-way ANOVA, diet, MG). The effect of maternal obesity on blood biochemical parameters was found only in males. SFD consumption was accompanied by an increase in cholesterol plasma levels only in males born to obese mothers (two-way ANOVA, diet, MG, *p* < 0.01 N-K). In addition, males born to obese mothers had higher levels of blood glucose than male offspring of control mothers when consuming the SFD (two-way ANOVA, diet, MG, *p* < 0.01 N-K).

Males and females did not differ in glucose tolerance, and SFD consumption reduced glucose tolerance in mice of both sexes: glucose concentrations and areas under curves (AUC) were higher in SFD-fed mice (*p* < 0.01, diet, repeated measures ANOVA for glucose concentrations; *p* < 0.01, diet, three-way ANOVA for AUC). Maternal obesity had no effect on glucose tolerance in the offspring, either on the SD or on the SFD (Figure 4).

SFD consumption was accompanied by an increase in the liver expression of the gene for FGF21, genes encoding lipogenesis enzymes fatty acid synthase (*Fasn*) and acetyl-coA carboxylase 1 (*Acaca*), the gene for apolipoprotein B (*ApoB*) involved in fat transport, and genes for glycolysis enzyme pyruvate kinase (*Pklr*) (Figure 5). *Fgf21* expression in mice consuming the SFD was almost twice as high in males than in females (Figure 5); the increase in the expression of other genes did not depend on sex. Maternal obesity increased liver expression of lipogenesis genes (*Fasn, Acaca,* and *Acacb*), and to a greater extent in males, only when consuming the SD (Figure 5). In mice that consumed the SFD, maternal obesity reduced liver expression of the *Ppara* gene.

In the hypothalamus, SFD consumption increased gene expression of anorexigenic factors (*Crh* and *Pomc*) and decreased gene expression of orexigenic factor *Npy* in mice of both sexes and had no effect on the expression of the genes for the leptin receptor and for MC4R (Figure 6). Gene expression of orexigenic factors (*Npy* and *Agrp*) in females was lower than in males. Maternal obesity did not affect the expression of the studied genes in the hypothalamus.

Thus, we did not find any effect of genetic maternal obesity on the susceptibility to diet-induced obesity and food choice in the offspring, but we found a sex-specific effect of maternal obesity on energy intake and expression of liver lipogenesis genes in offspring consuming SD and an effect on the liver *Ppara* expression in mice that consumed the SFD regardless of sex.

## 3. Discussion

Many factors contribute to the development of obesity, including maternal obesity and maternal malnutrition during pregnancy [13]. A significant proportion of reproductive-aged women are overweight now, and the number of pregnancies complicated by obesity increases constantly [27], which enhances the risk of obesity in the next generation. In this regard, recommendations are being developed for the use of various healthy diets for pregnant women [28], however, the programming effect of maternal obesity on the ability of offspring to develop obesity when overweight mothers consume a balanced diet remains unexplored.

In the present work, we investigated the influence of genetic maternal obesity, which develops when consuming a balanced diet, on the metabolic phenotype and susceptibility to diet-induced obesity in offspring. We assessed the response to SFD intake in offspring by morphometric, hormonal, metabolic, and transcriptional (in the liver and hypothalamus) changes. SFD consumption increased energy intake and was accompanied by the development of obesity and insulin resistance, as evidenced by the simultaneous increase in blood glucose and insulin levels and a decrease in glucose tolerance. Increased energy intake with the SFD was accompanied by the activation of gene expression of the anorexigenic neuropeptides POMC [29] and CRH [30] and inhibition (at the tendency level) of gene expression of the orexigenic [29] neuropeptide NPY. These transcriptional changes reflect a regulatory response of the hypothalamus aimed to downregulate energy intake. 

In offspring consuming an SFD, liver expression of genes for lipogenesis enzymes (*Fasn* and *Acaca*) and glycolysis (*Pklr*) increased, which, apparently, is a response to excessive consumption of sugars and an increase in glucose and insulin in the blood [31]; the expression of the apolipoprotein B gene also increased, which should increase the secretion of fats from the liver. In addition, SFD consumption sex-specifically upregulated the expression of the FGF21 gene, a hormone that adapts an organism to metabolic stress, increases insulin sensitivity, and reduces sugar intake [32]. In males, the diet-induced expression of the FGF21 gene in the liver was almost two times higher than in females, which is in line with our previous results [33].

We found that genetic maternal obesity did not affect either susceptibility to diet-induced obesity in the offspring or the development of insulin resistance, energy intake with SFD, and the transcriptional response of the studied genes in the hypothalamus. The only effect of maternal obesity on liver gene expression in SFD-eating offspring was reduction in *Ppara* expression. PPARa plays an important role in hepatic lipid metabolism, and downregulation of the expression of this gene in the liver of HFD-fed mice reduced fatty acid oxidation activity and promoted the development of NAFLD [34]. Although we did not find an effect of maternal obesity on liver weight, more research is required to answer the question of whether the inhibitory effect of maternal obesity on hepatic *Ppara* expression contributes to the development of NAFLD in mice.

Maternal obesity also did not have any significant effect on food choice in the offspring. Food choice was assessed under conditions of free choice between standard chow pellets, sweet fatty biscuits, and lard. The choice of food depends not only on taste preferences, but also on the tactile sensing of the texture of the food. Males consumed the same amount of SD pellets and biscuits, hard foods that are similar in texture, possibly indicating non-selectivity in choosing between these foods. Females reduced SD pellet consumption and preferred to consume sweet biscuits. As a result, females began to consume less standard food than males. One of the reasons for these differences may be sex differences in the functioning of AGRP neurons. AGRP neurons mediate the suppressive effect of HFD consumption on SD cravings [35], and we found an effect of sex on *Agrp* gene expression in the present work. In our experiment, when mothers consumed a standard diet, maternal obesity did not affect the choice of diet components in the offspring. These data confirm the assumption that maternal diet, not maternal obesity *per se*, has a programming effect on taste preferences in the offspring [36]. 

However, maternal obesity had a sex-specific programming effect on the phenotype of the offspring, which consumed a standard diet. As compared to males born to control mothers, the male offspring of obese mothers demonstrated a more wasteful type of metabolism, as they consumed more standard chow but did not differ in body weight. In addition, they had increased expression of lipogenesis enzyme genes in the liver, which may indicate an increase in the activity of liver lipogenesis [31]. This gene expression increase is possibly a consequence of the increased food intake in the offspring of obese mothers. Previously, we have shown that an increase in standard chow intake during refeeding was accompanied by an increase in *Fasn* gene expression in the liver [33].

Sex differences in developmental programming of energy intake and expenditure may be related to sexual dimorphism in the ontogenesis of systems regulating energy homeostasis [12]. The influence of sex steroids on neurogenesis in early life and in adulthood [12,37] may be a reason for these differences. Different placental responses to maternal obesity in male and female fetuses may also contribute to the sexual dimorphism in prenatal developmental programming [12]. We have previously shown that the *A^y^* mutation in mothers [38] and administrations of leptin to pregnant female mice [39] evoke different responses in the placentas of male and female fetuses.

Genetic obesity caused hyperinsulinemia in early pregnancy, hyperleptinemia and increased levels of FGF21 in late pregnancy, and reduced weights of placentas, fetuses, and newborns. Fetal and neonatal weights are integrating characteristics reflecting multiple prenatal influences, and maternal obesity can lead to increased, decreased, or unchanged placental and fetal weights depending on individual maternal metabolic disorders [40]. We and other authors have previously shown that hyperleptinemia during pregnancy is associated with reduced fetal and placental weights [38,39,41], which is consistent with the data obtained in the present study.

It is assumed that both leptin [42] and insulin [43] have a programming effect on the development of offspring. Obesity-associated hyperinsulinemia and hyperleptinemia in pregnant females could influence the programming of food intake regulation in the offspring under standard conditions but did not affect the offspring’s response to SFD intake. At the same time, numerous studies demonstrate the potentiating effect of maternal obesity caused by the consumption of high-calorie diets on the development of metabolic diseases in the offspring [13]. These discrepancies point to a significant role of maternal diet in programming the propensity to develop obesity in the offspring.

Not only the embryonic but also the early postnatal period of breastfeeding and maternal care represent an ontogenetic window for developmental programming [44]. During lactation, the differences between obese and control females significantly decreased (adiposity) or completely disappeared (hormones in the blood). Previously, we showed that pregnancy and lactation normalize the metabolism in *A^y^/a* females that were mated at the initial stages of obesity [45]. The results of this work show that pregnancy and lactation affect severely obese *A^y^/a* females in a similar normalizing manner. The normalization of metabolism in SD-fed *A^y^/a* females during lactation was combined with the absence of differences in weight and gene expression of IGF1 (a growth factor that affects growth and physical development after birth [46,47]) in the liver of their offspring. The absence of influence of genetic maternal obesity on the susceptibility to diet-induced obesity in the offspring may indicate that programming of offspring reaction to high-calorie intake occurs during early postnatal life and is consistent with the assumption that maternal high-calorie diets may have a greater influence on offspring outcome than maternal obesity [44].

In conclusion, genetic obesity in females during pregnancy is characterized by an altered hormonal background and has a sex-specific programming effect on energy intake and gene expression of lipogenesis enzymes in the liver when offspring consume a balanced diet, but does not affect the susceptibility to diet-induced obesity and food choice in offspring. These data, combined with literature data on the negative impact of diet-induced maternal obesity on taste preferences and risk of developing obesity in offspring, point to the importance of a healthy, balanced diet in overweight pregnant women.

## 4. Materials and Methods

### 4.1. Animals and Experimental Design

C57BL/6J mice with agouti genotypes *A^y^/a* and *a/a* were obtained in crosses *a/a* x *A^y^/a* in the vivarium of the Institute of Cytology and Genetics. The mice were housed under a 12:12 h light–dark regime (with lights switched off at 1800) at an ambient temperature of 22 °C. The mice were provided access to standard commercial mouse chow during maternal care and after weaning and water ad libitum. *A^y^/a* mice developed obesity with age consuming standard chow. At 18–20 wk. of age, when *A^y^/a* females became obese, females were mated with males in reciprocal crosses (*A^y^/a* × *a/a* and *a/a* × *A^y^/a*). The mating was judged from the presence of a copulation plug. The appearance of the plug signified day 0 of pregnancy. Mated females were housed individually and were monitored to record parturition and the number of pups, and the day of delivery was designated as postpartum day (PPD)1. Large litters (>7) were reduced to 7 on PPD1 by discarding pups of the lowest weights.

Experimental design is presented in Figure 7. In order to assess maternal metabolic characteristics during pregnancy and lactation, pregnant females were killed by rapid decapitation on pregnancy day (PD)5, PD17, and PPD10, and female trunk blood samples were collected. In pregnant females, subcutaneous and visceral fat tissue were dissected and weighed. On PD17, the weights of placentae and fetuses were measured. Female body weight on PD0, 5, and 17 and female and pup BW on PPD1 and PPD10 were measured. To measure liver gene expression in pups, the liver samples of *a/a* pups were snap-frozen in liquid nitrogen.

To assess maternal influence on offspring metabolism, 2 male or 2 female offspring of *a/a* genotype (normal metabolism) from each litter were separated from their mothers on PPD28 and then were housed individually with free access to water and standard diet until the age of 10 weeks. From the age of 10 weeks, one male and one female from each litter continued to receive SD, and one male and one female began to receive sweet fatty biscuits and lard in addition to the standard chow (sweet and fatty diet, SFD). This mixture mimics the cafeteria diet and potentiates the rapid development of obesity in mice [48]. The standard chow was replaced once a week, and biscuits and lard were replaced three times a week. The following parameters were recorded: the amounts of daily eaten standard chow, biscuits, and lard; the amounts of energy consumed with these kinds of food (lard, 8 kcal/g; standard chow, 2.5 kcal/g; biscuits, 4.58 kcal/g); and the energy consumed with each kind of food as a percentage of the entire energy consumed. In the last week of the experiment, the mice were subjected to a glucose tolerance test (GTT) and then were sacrificed by decapitation three days after the test. Samples of blood were collected, the weights of the liver, subcutaneous, and abdominal fat were measured, and samples of hypothalamus and liver were excised and immediately snap-frozen in liquid nitrogen.

### 4.2. Glucose Tolerance Test

Mice were fasted in the morning for 6 hours prior to glucose delivery via intraperitoneal injection at a dose of 1 g/kg. Blood was sampled from the tail vein before the injection (0 min) and then at 15, 30, 60, and 120 min after glucose injection. Blood glucose concentrations were determined with a glucometer (One Touch Select Plus, LifeScan Johnson&Johnson, LifeScan Europe, a Division of Cilag GmbH International, Gubelstrasse 34, CH-6300, Zug, Switzerland).

### 4.3. Diets 

Standard chow was purchased from BioPro, Novosibirsk, Russia. Composition: two-component grain mixture, milk components, high-protein components (vegetable and animal proteins), vegetable oil, amino acids, organic acids, vitamin-mineral premix, and fiber. Crude protein: 22%. Energy value 2500 kcal. Pork lard and biscuits were bought in a food store. Biscuits composition (g/100 g): proteins—6.9, fats—18.4, and carbohydrates—71.8. Energy value 458 kcal/100 g. Lard (subcutaneous fat): proteins—1.8, fats—94.2, and carbohydrates—0. Energy value 800 kcal/100 g. 

### 4.4. Plasma Assays and Glycogen Measurements

Concentrations of insulin and leptin were measured using Rat/Mouse Insulin ELISA Kit and Mouse Leptin ELISA Kit (EMD Millipore, St. Charles, MO, USA). Concentrations of glucose, triglycerides, and cholesterol were measured colorimetrically using Fluitest GLU, Fluitest TG, and Fluitest CHOL (Analyticon^®^ Biotechnologies AG Am Mühlenberg 10, 35,104 Lichtenfels, Germany), respectively. Liver glycogen was measured by the method of Roehrig and Allred [49].

### 4.5. Relative Quantitation Real-Time PCR

Total RNA was isolated from tissue samples using ExtractRNA kit (Evrogen, Moscow, Russia) according to the manufacturer’s instructions. First-strand cDNA was synthesized using Moloney murine leukemia virus (MMLV) reverse transcriptase (Evrogen, Moscow, Russia) and oligo(dT) as a primer. TaqMan gene expression assays (Thermo Fisher Scientific, Waltham, MA USA) were used for relative quantitative real-time PCR with β-actin (Actb) as an endogenous control: acetyl-coenzyme A carboxylase alpha, Acca, Mm01304285_m1; acetyl-coenzyme A carboxylase beta, Accb, Mm01204683_m1; agouti related neuropeptide, Agrp, Mm00475829_g1; apolipoprotein B, Apob, Mm01545150_m1; beta-actin, Actb, Mm00607939_s1; carnitine palmitoyltransferase 1a, Cpt1a, Mm01231183_m1; corticotropin releasing hormone, Crh, Mm01293920_s1; fatty acid synthase, Fasn, Mm00662319_m1; fibroblast growth factor 21, Fgf21, Mm00840165_g1; glucokinase, Gck, Mm00439129_m1; leptin receptor, Lepr, Mm00440181_m1; melanocortin receptor type 4, Mc4r, Mm00457483_s1; neuropeptide Y, Npy, Mm01410146_m1; peroxisome proliferator activated receptor alpha, Ppara, Mm0040939_m1; pro-opiomelanocortin, Pomc, Mm00435874_m1; and pyruvate kinase liver and red blood cell, Pklr, Mm00443090_m1. Sequence amplification and fluorescence detection were performed on an Applied Biosystems ViiA 7 Real-Time PCR System (Life Technologies, 5791 Van Allen Way, Carlsbad, CA 92008 USA). Relative quantification was performed by the comparative threshold cycle (CT) method.

### 4.6. Statistical Analysis

The results are presented as means ± SE from the indicated number of mice. Two-way ANOVA was used to compare biochemical characteristics of obese *(A^y^/a)* and control (*a/a*) mothers with factors “genotype” (*a/a*, *Ay/a*) and “day” (PD5, PD17, and PPD10). To examine the maternal influence on the response to SFD consumption, offspring morphometric characteristics, plasma biochemical characteristics, and gene expressions in the liver and hypothalamus were analyzed initially by three-way ANOVA with the factors “maternal genotype” (MG: *a/a*, *A^y^/a*), “sex” (male, female), and “diet” (SD, SFD), and then separately by two-way ANOVA in males and females. To examine the maternal influence on the food choice, food intake and share of energy consumed with different types of food were analyzed initially by three-way ANOVA with the factors “MG”, “sex,” and “type of food” (chow, biscuits, and lard) and then separately by two-way ANOVA for different types of food. Repeated measures ANOVA was used to compare offspring BW and food intake after weaning at the age of 4–10 weeks with factors “MG” and “sex”, and at the age of 10–20 weeks with factors “MG”, “sex”, and “diet”, and to analyze the results of GTT with factors “MG”, “sex”, and “minutes” (0, 15, 30, 60, and 120). In addition, multiple comparisons were performed with the post hoc Newman–Keuls test. The comparisons between single parameters were performed with a two-tailed Student’s *t*-test. Significance was determined as *p* < 0.05. The STATISTICA 6 software package (StatSoft, TIBCO Software Inc., Palo Alto, CA, USA) was used for analysis.

## Figures and Tables

**Figure 1 ijms-24-05610-f001:**
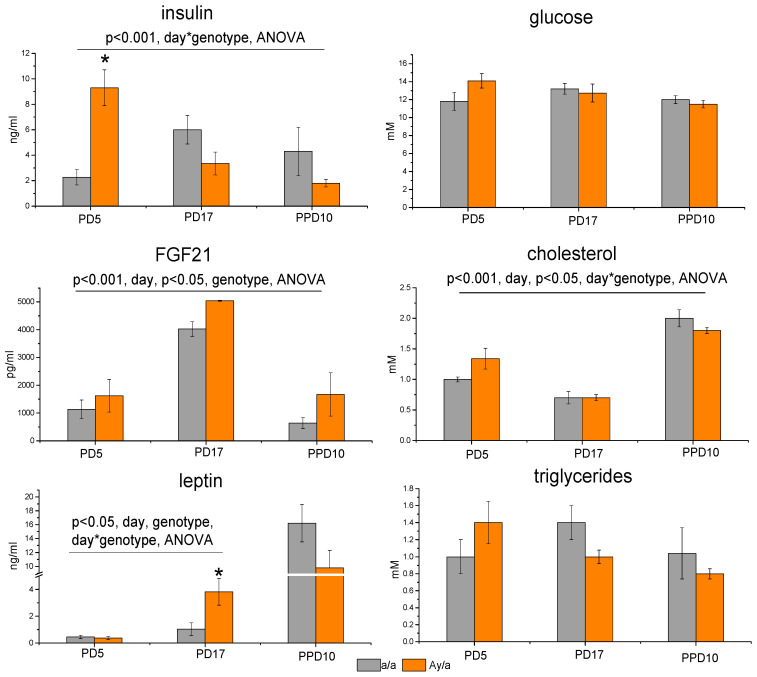
Hormonal and metabolic characteristics of females during pregnancy and lactation: * *p* < 0.05; post hoc Newman–Keuls test between *a/a* and *A^y^/a*; PD5,17—pregnancy day 5, 17; PPD10—postpartum day 10. Data are presented as mean ± SE for 7 mice in each group. Data were analyzed with two-way ANOVA with factors “day” (PD5, PD17, and PPD10) and “genotype” (*a/a* and *A^y^/a*).

**Figure 2 ijms-24-05610-f002:**
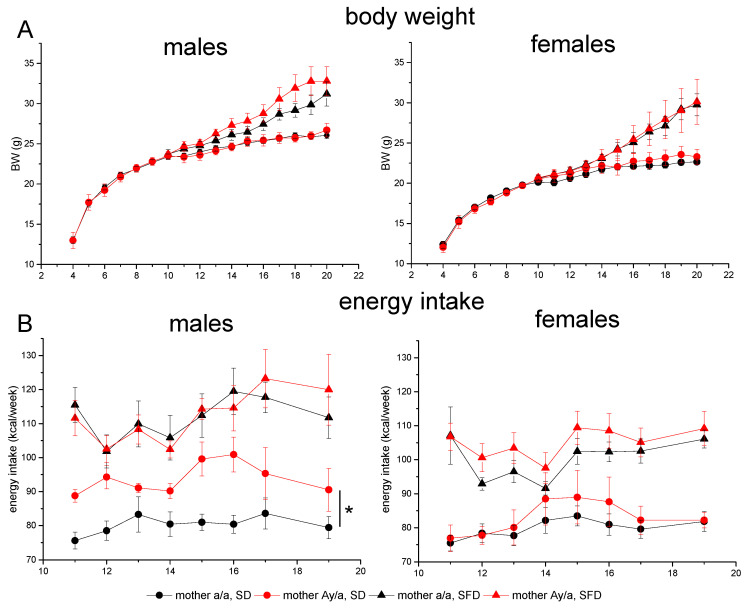
Influence of maternal obesity on body weight (**A**) and weekly energy intake (**B**) in male and female offspring consuming SD or SFD diets. Male and female offspring of obese (*A^y^/a*) and control (*a/a*) mothers were fed with SD or SFD for 10 weeks from the age of 10 weeks. Data are presented as mean ± SE. The number of animals in groups is shown in Table 2. Data were analyzed by repeated measures ANOVA with factors “sex”, “diet” and “maternal genotype (MG)” for all groups and separately for males and females that consumed different diets with factor “MG”: * *p* < 0.05; MG, repeated measures ANOVA for SD-males. Axis X designates mouse age (weeks).

**Figure 3 ijms-24-05610-f003:**
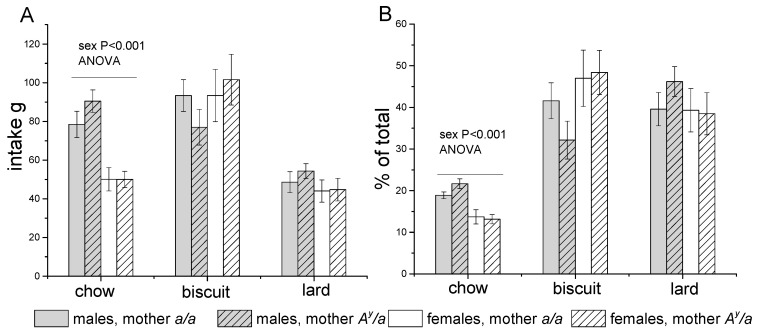
Total amount of food (g, (**A**)) and share of energy (% of total, (**B**)) consumed with different diet components by males and females born to *a/a* and *A^y^/a* mothers during 10 weeks of diet consumption. Data are presented as mean ± SE for 6–8 mice in each group. Data were analyzed by two-way ANOVA with factors “sex” and “MG” for each component of diet.

**Figure 4 ijms-24-05610-f004:**
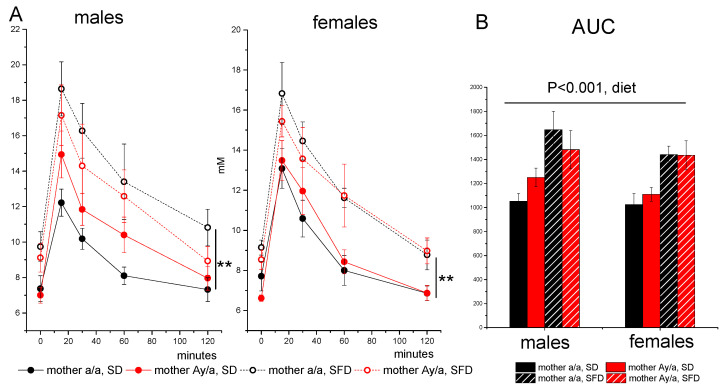
Blood glucose concentrations (**A**) and areas under curves (**B**) during i.p. glucose tolerance test in male and female offspring of control (*a/a*) and obese (*A^y^/a*) mothers. Male and female offspring were fed with SD or SFD during 10 weeks from the age of 10 weeks. Data are presented as mean ± SE. The number of animals in groups is shown in Table 2. Data were analyzed by repeated measures ANOVA with factors “sex”, “diet”, and “MG” for five repeated measures of glucose in blood, and by three-way ANOVA with factors “sex”, ‘diet”, and “MG” for AUC: ** *p* < 0.01, diet, repeated measures ANOVA.

**Figure 5 ijms-24-05610-f005:**
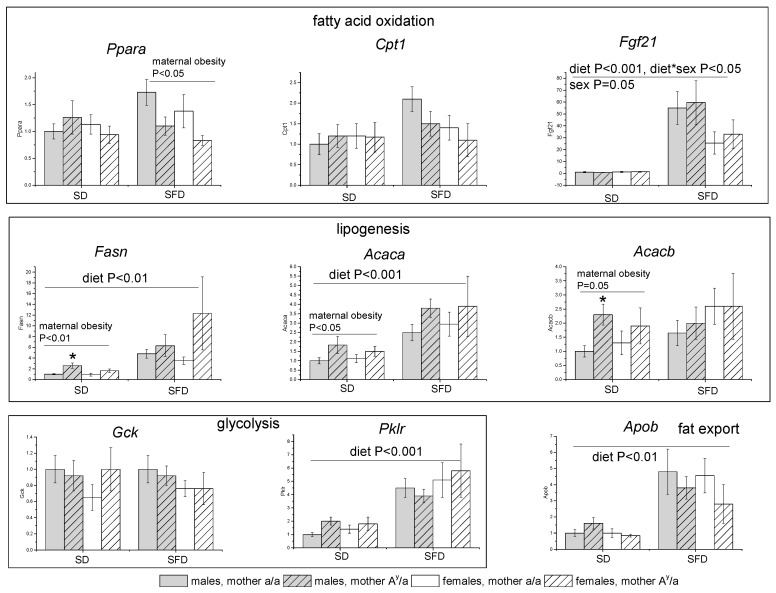
Liver gene expression in male and female offspring of control and obese mothers under consumption of SD or SFD. Male and female offspring of obese (*A^y^/a*) and control (*a/a*) mothers were fed with SD or SFD for 10 weeks from the age of 10 weeks. Data are presented as mean ± SE for 5–7 mice in each group. Y axis designates fold changes in gene expression as compared to group of SD males born to control (*a/a*) mothers. Data were analyzed by three-way ANOVA with factors “sex”, “diet”, and “MG” and separately for SD and SFD by two-way ANOVA with factors “sex” and “MG”. * *p* < 0.05, post hoc Newman–Keuls test between male offspring of *a/a* and *A^y^/a* mothers.

**Figure 6 ijms-24-05610-f006:**
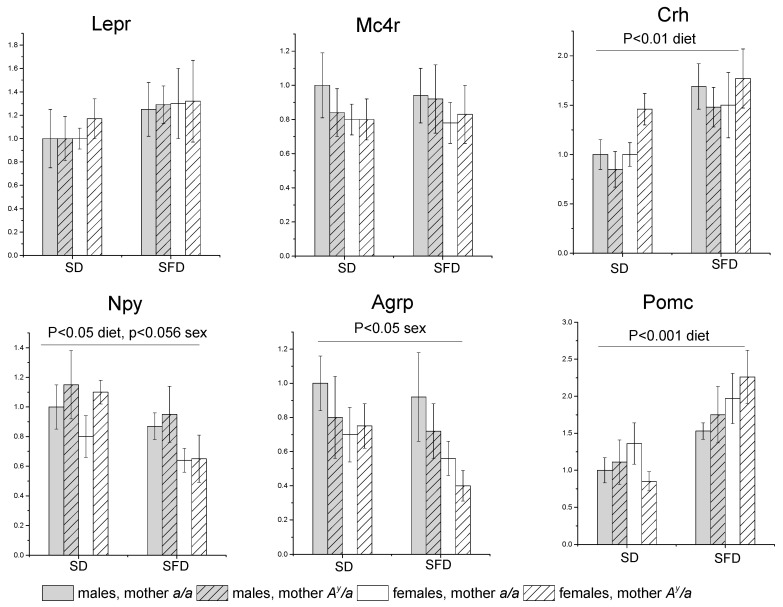
Hypothalamic gene expression in male and female offspring of control and obese mothers under consumption of SD or SFD. Male and female offspring of obese (*A^y^/a*) and control (*a/a*) mothers were fed with SD or SFD for 10 weeks from the age of 10 weeks. Data are presented as mean ± SE for 5–7 mice in each group. Y axis designates fold changes in gene expression as compared to group of SD males born to control (*a/a*) mothers. Data were analyzed by three-way ANOVA with factors “sex”, “diet”, and “MG”.

**Figure 7 ijms-24-05610-f007:**
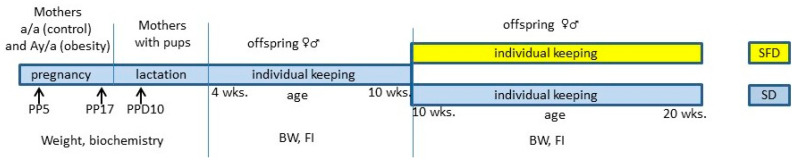
Experimental design. The top line shows the mice under study, and the bottom line shows the measured parameters. Blue color indicates the period of consumption of the standard diet; yellow color—the period of consumption of the sweet and fatty diet. There were 6 experimental groups of mothers (*A^y^/a* and *a/a* mothers at PP5, PP17, and PPD10) with 7 females in each group, and 8 experimental groups of the offspring: 6 male and 8 female offspring of *A^y^/a* mothers and 7 male and 7 female offspring of *a/a* mothers that consumed SFD; 5 male and 8 female offspring of *A^y^/a* mothers and 7 male and 8 female offspring of *a/a* mothers that consumed SD. FI – food intake.

**Table 1 ijms-24-05610-t001:** Weight characteristics of females and their offspring during pregnancy and lactation.

		*a/a*			*A^y^/a*		
		N	Mean	SE	N	Mean	SE
Pregnancy day 0	Female BW (g)	15	22.4	0.3	14	32.6 *	0.5
Pregnancy day 5	Female BW (g)	7	23.2	0.6	7	34.3 *	1.6
	Fat weight (g)	7	1.24	0.11	7	7.77 *	0.85
Pregnancy day 17	Female BW (g)	7	34.5	0.42	7	41.3 *	1.4
	Fat weight (g)	7	2.15	0.2	7	6.95 *	0.57
	Placenta weight (mg)	46	98.3	2.7	56	90 *	1.9
	Fetal weight (mg)	45	814	18.5	57	687 *	11
Postpartum day 0	Female weight (g)	7	25.8	0.86	6	30 *	0.7
	Pup weight (g)	51	1.3	0.015	55	1.21 *	0.016
Postpartum day 10	Female weight (g)	7	26	0.7	7	29.3 *	0.44
	Pup weight (g)	43	4.1	0.08	55	4.26	0.07

* *p* < 0.05 Student *t*-test between *a/a* and *A^y^/a* female mice. BW—body weight.

**Table 2 ijms-24-05610-t002:** Weight characteristics in male and female offspring of control and obese mothers on a standard diet and sweet and fatty diet.

	Males				Females				
	SD		SFD		SD		SFD		P ANOVA
	Mother *a/a*	Mother *A^y^/a*	Mother *a/a*	Mother *A^y^/a*	Mother *a/a*	Mother *A^y^/a*	Mother *a/a*	Mother *A^y^/a*	
N	7	5	7	6	8	8	7	8	
weight (g)									
liver	1.29 ± 0.04	1.29 ± 0.10	1.29 ± 0.13	1.31 ± 0.05	1.09 ± 0.03	1.16 ± 0.05	1.20 ± 0.05	1.19 ± 0.01	sex *p* < 0.01
abdominal fat	0.69 ± 0.24	0.31 ± 0.07	1.70 ± 0.30	2.15 ± 0.42	0.29 ± 0.09	0.41 ± 0.05	2.38 ± 0.39	1.96 ± 0.41	diet *p* < 0.001
subcutaneous fat	0.69 ± 0.21	0.56 ± 0.09	2.07 ± 0.47	2.54 ± 0.47	0.53 ± 0.07	0.65 ± 0.09	3.08 ± 0.46	3.06 ± 0.65	diet *p* < 0.001
total fat	1.38 ± 0.41	0.87 ± 0.15	3.77 ± 0.75	4.70 ± 0.75	0.82 ± 0.14	1.06 ± 0.12	5.46 ± 0.84	5.02 ± 0.93	diet *p* < 0.001
index (%)									
liver	4.94 ± 0.14	4.80 ± 0.25	4.17 ± 0.47	4.01 ± 0.12	4.82 ± 0.09	4.98 ± 0.01	4.03 ± 0.01	4.06 ± 0.02	diet *p* < 0.001
abdominal fat	2.65 ± 0.94	1.14 ± 0.25	5.28 ± 0.67	6.38 ± 1.03	1.29 ± 0.42	1.75 ± 0.20	7.76 ± 1.03	6.22 ± 0.96	diet *p* < 0.001
subcutaneous fat	2.65 ± 0.83	2.06 ± 0.28	6.38 ± 1.07	7.54 ± 1.20	2.33 ± 0.27	2.73 ± 0.32	10.06 ± 1.13	9.50 ± 1.09	sex *p* < 0.05, diet *p* < 0.001, sex * diet *p* < 0.05
total fat	5.31 ± 1.61	3.20 ± 0.45	11.66 ± 1.67	13.93 ± 2.17	3.62 ± 0.60	4.49 ± 0.40	17.82 ± 2.12	15.71 ± 1.61	diet *p* < 0.001, sex * diet *p* = 0.05

**Table 3 ijms-24-05610-t003:** Hormonal and metabolic characteristics in male and female offspring of control and obese mothers on a standard diet and sweet and fatty diet.

	Males				Females				
	SD		SFD		SD		SFD		P ANOVA
	Mother *a/a*	Mother *A^y^/a*	Mother *a/a*	Mother *A^y^/a*	Mother *a/a*	Mother *A^y^/a*	Mother *a/a*	Mother *A^y^/a*	
N	6	5	7	6	7	6	7	7	
FGF21 (ng/mL)	0.43 ± 0.14	0.17 ± 0.02	8.83 ± 3.33	11.52 ± 1.73	0.33 ± 0.06	0.29 ± 0.03	7.23 ± 2.28	8.18 ± 2.26	diet *p* < 0.001
insulin (ng/mL)	3.33 ± 0.64	4.10 ± 0.62	5.79 ± 0.87	7.41 ± 1.15	3.10 ± 0.56	3.38 ± 1.05	6.49 ± 1.07	4.50 ± 0.90	diet *p* < 0.001
glucose (mM)	8.74 ± 0.64	9.04 ± 0.45	12.42 ± 0.70	15.13 ± 0.60	11.03 ± 0.82	9.52 ± 0.52	15.09 ± 0.68	13.86 ± 0.66	diet *p* < 0.001, sex *p* < 0.05, sex * MG
fasting glucose (mM)	7.37 ± 0.73	7.00 ± 0.44	9.74 ± 0.84	9.11 ± 0.81	7.71 ± 0.74	6.61 ± 0.16	9.14 ± 0.37	8.54 ± 0.47	diet *p* < 0.001
cholesterol (mM)	2.18 ± 0.12	1.73 ± 0.16	2.07 ± 0.14	2.49 ± 0.14	2.23 ± 0.31	2.16 ± 0.41	2.04 ± 0.09	1.86 ± 0.07	NS
triglycerides (mM)	1.29 ± 0.15	1.01 ± 0.11	0.94 ± 0.08	1.33 ± 0.14	1.37 ± 0.20	1.11 ± 0.13	1.09 ± 0.10	1.08 ± 0.13	diet * MG *p* < 0.05
liver glycogen (mg/g)	18.9 ± 2.1	21.8 ± 2.6	22.6 ± 2.6	26.7 ± 2.7	20.3 ± 1.6	22.7 ± 2.0	30.8 ± 2.4	29.8 ± 4.5	diet *p* < 0.001

## Data Availability

The data presented in this study are available on request from the corresponding author.

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
