# Peer review of "Genetic Obesity in Pregnant Ay Mice Does Not Affect Susceptibility to Obesity and Food Choice in Offspring"

_ijms, 2023, doi:10.3390/ijms24065610_

Round 1

Reviewer 1 Report

Introduction

- Check line 49

The idea is good, but the authors need to improve.

Materials and methods

The main issue is that it is not clear who eats the control and cafeteria diets, the mothers or the pups?

It is suggested to review the experimental design and rewrite

Results

The graphs and tables need to improve, table 1 includes commas and decimal points.

Figure 5. y-axis what is the unit? relative expression?

Figure 6. Units

All figure captions must include, title, abbreviations used, the n that is considered in the experiment shown. What statistical analysis was used and the p values.

Discussion

Review the idea lines 243-245

Author Response

Introduction

- Check line 49

The idea is good, but the authors need to improve.

Thank you very much, it was mistake, refers have been changed

Materials and methods

The main issue is that it is not clear who eats the control and cafeteria diets, the mothers or the pups?

It is suggested to review the experimental design and rewrite

We have corrected the Materials and Methods section and added a picture showing the experiments performed.

Results

The graphs and tables need to improve, table 1 includes commas and decimal points.

It was done

Figure 5. y-axis what is the unit? relative expression?

It was indicated

Figure 6. Units

All figure captions must include, title, abbreviations used, the n that is considered in the experiment shown. What statistical analysis was used and the p values.

It was done, figure captions were corrected

Discussion

Review the idea lines 243-245

Discussion was rewritten

Author Response

 The study explored an interesting question that is programming effect of maternal obesity by virtue of genetics on the ability of offspring to develop obesity when overweight mothers consume a balanced diet. Overall, the study findings are interesting and give an important insight into how the balanced diet consumed by pregnant mothers can neutralize the harmful genetic effects on boosting diet induced obesity. However, there are some of the key questions that needs to be addressed in the revised manuscript as follows:

  1. It is interesting to observe that placenta weight and fetal weight is low in mutant Ay/a females than normal a/a females although Ay/a females consistently have high body weight than normal ones throughout their pregnancy (Table1). A detailed justification is needed.

It was done in the section Duscussion:

 Genetic obesity caused hyperinsulinemia in early pregnancy, and hyperleptinemia and increased levels of FGF21 in late pregnancy, and reduced weights of placentas, fetuses and newborns. Fetal and neonatal weight is an integrating characteristic reflecting multiple prenatal influences, and maternal obesity can lead to increased, decreased, or unchanged placental and fetal weights depending on individual maternal metabolic disorders [40]. We and other authors have previously shown that hyperleptinemia during pregnancy is associated with reduced fetal and placental weight [38,39,41] , which is consistent with the data obtained in the present study.

  1. One of the key and useful observation from the study is that when obese mothers consume a balanced diet, mother’s obesity status did not influence food choice and development of diet-induced obesity in offspring. However, the timing of diet given to mothers is not clear. When were mother mice put on standard diet vs. SFD diet? Was it immediately after their birth or during puberty or few days/weeks/months before mating?

Mice with mutation Ay develop obesity consuming balanced diet   because overeating (they consume about 25% more food than control a/a mice) and decreased energy expenditure. Obesity in Ay mice develops with age. Ay/a mothers consumed only SD during all their life, however, we began to mate them only at the age 18-20 weeks when they became obese. We indicate this in rewritten section MM and include figure demonstrating design of experiment.

  1. Lipogenesis genes such as (Fasn, Acaca and Acacb) were significantly elevated in male offsprings born to mothers with Ay/a mutation even when they were on standard diet indicating major influence of genetic mutation passed by mother on offspring’s lipogenesis processes that is independent of mother’s diet (Figure 5). A discussion is needed as to why this effect was specifically observed in males and not females.

We included this in Discussion:

Sex differences in developmental programming of energy intake and expenditure may be related to sexual dimorphism in ontogenesis of systems regulating energy homeostasis [12]].  The influence of sex steroids on neurogenesis in early life and in adulthood [12,37]  may be a reason of these differences. Different placental responses to maternal obesity in male and female fetuses may also contribute to the sexual dimorphism in prenatal developmental programming [12]. We have previously shown that Ay mutation in mothers [38] and administrations of leptin to pregnant female mice [39] evoke different response in the placentas of male and female fetuses.

  1. Authors measured only leptin, which is one of the adipocytokine. Obesity is inflammatory condition and therefore it is important to study various other adipokines and inflammatory markers such as resistin, adiponectin, IL-6, IL-12, CRP etc. besides leptin It would be interesting to have these markers profiles in studied comparisons if feasible.

Of course, other adipokines, not only leptin, but also pro-inflammatory cytokines, may have a programming effect. Unfortunately, we did not measure the concentrations of pro-inflammatory cytokines in the blood. We focused on leptin because we had previously shown that the administration of leptin to pregnant female mice has a programming effect on the metabolic phenotype of the offspring (doi:10.1002/2211-5463.12757).

  1. A graphical abstract of the study is highly encouraged to make readers get the most from the article.

Of course, we agree, and now we are working on a graphic abstract.

Round 2

Reviewer 2 Report

Graphical abstract is not available in the revised version. Graphical abstract is recommended before final publication.